# Renal Manifestations of Covid-19: Physiology and Pathophysiology

**DOI:** 10.3390/jcm10061216

**Published:** 2021-03-15

**Authors:** Zaher Armaly, Safa Kinaneh, Karl Skorecki

**Affiliations:** 1Department of Nephrology, Nazareth Hospital, EMMS, Nazareth 16100, Israel; dialysis_research3@nazhosp.com; 2The Bar-Ilan University Azrieli Faculty of Medicine, Safed 1311502, Israel; karl.skorecki@biu.ac.il

**Keywords:** COVID19 pandemic, angiotensin converting enzyme 2, SARS-CoV-2, kidney, acute kidney injury, dialysis, treatment

## Abstract

Corona virus disease 2019 (COVID-19) imposes a serious public health pandemic affecting the whole world, as it is spreading exponentially. Besides its high infectivity, SARS-CoV-2 causes multiple serious derangements, where the most prominent is severe acute respiratory syndrome as well as multiple organ dysfunction including heart and kidney injury. While the deleterious impact of SARS-CoV-2 on pulmonary and cardiac systems have attracted remarkable attention, the adverse effects of this virus on the renal system is still underestimated. Kidney susceptibility to SARS-CoV-2 infection is determined by the presence of angiotensin-converting enzyme 2 (ACE2) receptor which is used as port of the viral entry into targeted cells, tissue tropism, pathogenicity and subsequent viral replication. The SARS-CoV-2 cellular entry receptor, ACE2, is widely expressed in proximal epithelial cells, vascular endothelial and smooth muscle cells and podocytes, where it supports kidney integrity and function via the enzymatic production of Angiotensin 1-7 (Ang 1-7), which exerts vasodilatory, anti-inflammatory, antifibrotic and diuretic/natriuretic actions via activation of the Mas receptor axis. Loss of this activity constitutes the potential basis for the renal damage that occurs in COVID-19 patients. Indeed, several studies in a small sample of COVID-19 patients revealed relatively high incidence of acute kidney injury (AKI) among them. Although SARS-CoV-1 -induced AKI was attributed to multiorgan failure and cytokine release syndrome, as the virus was not detectable in the renal tissue of infected patients, SARS-CoV-2 antigens were detected in kidney tubules, suggesting that SARS-CoV-2 infects the human kidney directly, and eventually induces AKI characterized with high morbidity and mortality. The mechanisms underlying this phenomenon are largely unknown. However, the fact that ACE2 plays a crucial role against renal injury, the deprivation of the kidney of this advantageous enzyme, along with local viral replication, probably plays a central role. The current review focuses on the critical role of ACE2 in renal physiology, its involvement in the development of kidney injury during SARS-CoV-2 infection, renal manifestations and therapeutic options. The latter includes exogenous administration of Ang (1-7) as an appealing option, given the high incidence of AKI in this ACE2-depleted disorder, and the benefits of ACE2/Ang1-7 including vasodilation, diuresis, natriuresis, attenuation of inflammation, oxidative stress, cell proliferation, apoptosis and coagulation.

## 1. Introduction

The renin angiotensin aldosterone system (RAAS) is a cardinal endocrine system that plays a key role in the regulation of blood pressure, fluid homeostasis, renal function and cardiac performance [1,2]. The classical RAAS consists of the protease renin, which is secreted from renal juxtaglomerular cells localized to the afferent arteriole and acts on the circulating precursor angiotensinogen to generate angiotensin (Ang) I, an inactive 10 amino acid (aa) peptide. The latter is converted by angiotensin-converting enzyme (ACE) to Ang II, an 8 aa active peptide. Ang II is the main effector component of the RAAS, as evident by its potent vasoconstrictor, pro-inflammatory and profibrotic actions (Figure 1A). Considering its deleterious contribution to cardiovascular and renal diseases, ACE inhibitors (ACEi) and Ang II receptor blockers (ARBs) have been developed and are very widely used in patients with hypertension, heart failure and chronic kidney disease.

Despite many decades of investigation, study of the RAAS keeps evolving and generating new knowledge. Specifically, new components of the RAAS system were discovered, namely ACE2, Ang (1-7) and G protein-coupled receptor, Mas, as an Ang (1-7) binding site (MasR) [1,3,4,5,6,7] (Figure 1). ACE2 was initially discovered in the testis, kidney and heart. However subsequent studies demonstrated wide abundance of the enzyme in the lung, liver, small intestine and brain [8,9,10]. ACE2 is partitioned between circulating and membrane components, with the vast majority being in the form a single pass transmembrane protein whose extracellular ectodomain mediates the catalytic cleavage activity, which converts Ang I into Ang (1–9) and, in turn, to Ang (1-7) by ACE [1,4,5,6,7].

In addition, ACE2 can directly convert Ang II to Ang 1-7. Importantly, ACE2 has a 400-fold affinity for Ang II as compared with the classic ACE. In contrast to Ang II, Ang 1-7 exerts vasodilatory, antiproliferative and apoptotic activity via its specific MasR, thus counterbalancing the adverse effects of Ang II [1,4,5,7].

As was already shown for SARS-Cov-1, the SARS-Cov-2 viral spike glycoprotein also binds with high affinity to ACE2, where it triggers its internalization along with the virus [11,12,13,14]. This might be of cardinal importance for pulmonary, cardiac and renal cells of infected subjects, especially patients with heart failure, diabetes, pulmonary diseases and hypertension, in which ACE2 levels are upregulated [15,16]. In a vicious feed-forward cycle, occupancy of, and intracellular translocation of SARS-CoV-2 coupled with ACE2 leads to its depletion, loss of its endogenous normal enzymatic function and concomitant depletion of its catalytic product, Ang (1-7), resulting in elimination of its beneficial effects (Figure 1B). Ang (1-7) activity, acting through its cognate Mas receptor, is necessary for maintaining the integrity of those crucial cell types and organ systems that are known to be adversely affected in the pathophysiology of COVID-19 disease. These are, most prominently, the lung alveolar cells, but also the cardiovascular, gastrointestinal and renal systems. The current review focuses on the physiology of ACE-2/Ang1-7/MasR axis and the renal manifestations of COVID 19.

## 2. Angiotensin Converting Enzyme 2 and Kidney Function

The kidney is both the source and target organ of the classic RAAS, as it contains all the components of this axis including renin, ACE, Ang II, AT-1 and AT2 receptors (AT1R and AT2R) [2]. Moreover, the kidney is among the organs richest in ACE2, where it is abundantly expressed in the various cell types but preferentially in the brush border of proximal tubular cells and, to a lesser extent, in podocytes and vascular endothelial cells, but not in glomerular endothelial and mesangial cells [4,17,18]. MasRs are mainly localized to the proximal and distal tubules but are also present in the glomerulus [19]. Interestingly, the local intra-renal concentrations of Ang (1-7) are comparable to those of Ang II and thought to play an import role in the regulation of kidney function and renal hemodynamics [6,20]. In this context, administration of Ang (1-7) into anesthetized rats enhances renal blood flow (RBF) [21], probably secondary to afferent arteriole vasodilation mediated by an NO-dependent mechanism [22]. Furthermore, accumulating evidence suggests that Ang (1-7) possesses diuretic and natriuretic actions, which may involve hemodynamic alterations and inhibitory effects of the reabsorption of sodium in the proximal tubule [6,23]. In contrast, other studies have demonstrated antinatriuretic/diuretic effects of Ang (1-7) in animals [24], most likely due to stimulation of water transport in the rat inner medullary collecting duct involving vasopressin V2 receptors [24,25]. Support for this notion was derived from the observation that administration of either acute or chronic A-779 (Ang (1-7) antagonist) into rats provoked natriuresis and diuresis responses [24,26,27]. The presence of Mas mRNA and immunoreactive protein in the proximal tubular cells concomitantly with abundant ACE2 [28], suggest that locally generated Ang (1-7) exerts advantageous tubular and hemodynamic effects, and probably a renal protective role, as it inhibits Ang II-induced phosphorylation of mitogen-activated protein (MAP) kinases [28,29]. In agreement with this assumption, Mas knockout mice had significant reductions in urine volume and fractional sodium excretion, and microalbuminuria concomitant with an exaggerated estimated glomerular filtration rate (eGRF) along a reduced renal blood flow, suggesting that glomerular hyperfiltration occurs in MasR-deficient mice [30]. Histological analysis revealed reduced glomerular tuft diameter and increased expression of collagen IV and fibronectin in both the mesangium and interstitium, associated with increased collagen III in the interstitium [30]. However, not all studies evidenced positive actions of Ang (1-7) [31]. For instance, renal deficiency for MasR diminished renal damage in models of renal insufficiency as unilateral ureteral obstruction and ischemia/reperfusion injury, while the infusion of Ang (1-7) to wild-type mice pronounced the pathological outcome by aggravating the inflammatory response [32]. It should be emphasized that these finding are at odds with the bulk of research results, which report beneficial effects of ACE2/Ang 1-7/MasR in determining the balance between the intrarenal effects of angiotensin II and angiotensin (1-7), thus rendering this axis a target for novel therapeutic approaches in a variety of kidney disorders [33].

## 3. Angiotensin Converting Enzyme 2 and Kidney Diseases

Changes in renal ACE2 distribution have been documented in various kidney diseases. In this context, both experimental and clinical diabetes and hypertension have been shown to be associated with alterations in ACE2 abundance/expression in the renal tissue. Clinical studies on human renal specimens revealed decreased expression of ACE2 in the glomerular endothelium and mesangial cells of diabetic patients [18]; but not in type 2 diabetic renal samples [34]. An additional clinical study on renal biopsies revealed that diabetic patients exhibited decreased ACE2 and increased ACE in tubulointerstitium and glomeruli as compared to healthy controls [35], suggesting that the kidney, like the heart, is vulnerable to injury upon reduced ACE2.

Noteworthy is that experimental studies applying models of type 2 diabetes have shown at early stages, prior to diabetic nephropathy development, that ACE2 expression is reduced in the kidney, while ACE expression is elevated [17]. In this context, a db/db mice model of type II diabetes exhibited reduction in glomerular expression of ACE2 along with upregulation of this enzyme in the renal cortex [17]. Likewise, streptozotocin-induced diabetes in mice resulted in significant elevation in ACE2 expression in the early stage of diabetic nephropathy [36] and decreased in late stage of the disease [37]. In order to explore the involvement these changes in the pathogenesis of diabetic nephropathy, MLN-4760 (a blocker of ACE2) was chronically (4–16 weeks) administered to animals with type I [17] and type II diabetes mellitus (DM) [38]. This pharmacological maneuver resulted in the development of proteinuria (3–5-fold increase). The development of the proteinuria was associated with expansion of the glomerular matrix in diabetic mice treated with MLN-4760 as compared to vehicle-treated mice. These observations could be explained by the findings of Oudit et al. [39] who reported early accumulation of fibrillar collagen (Collagen I and III) and fibronectin in the glomerular mesangial cells of male ACE2 mutant (ACE2−/y) mice, followed by development of glomerulosclerosis at 12 months of age, whereas female ACE2 mutant (ACE2−/−) mice were relatively protected. The vulnerable male mice also exhibited increased urinary albumin excretion and lipid peroxidation as compared to their age-control group. The deleterious renal consequences of ACE2 manipulation were ameliorated by the AT-1 receptor blocker, irbesartan. In agreement with this concept, Ace2(-/-) mice developed a more severe diabetic kidney injury (evident by albuminuria and glomerular damage) as compared with controls that underwent similar induction of the disease [40]. Moreover, infusion of Ang (1-7) ameliorated diabetic renal damage in an experimental model in both mice and rats [41,42,43]. Similarly, AVE0991 (MasR agonists) exerted nephroprotective effects against an ischemia/reperfusion murine model [44]. Of notice is that the targeted disruption of the ACE2 gene in mice was associated with the development of hypertension and exaggerated accumulation of Ang II in the kidney [45]. In line with these findings, spontaneously hypertensive rats (SHR) displayed aberrant renal ACE2 expression corresponding to the developmental pattern with declining expression during development and onset of hypertension [46]. Treatment with RAAS blockers (ACE-I or ARBs) upregulated ACE2 in the renal tissue of Lew. Tg (mRen2) congenic rats [47] similar to the impact of these drugs on cardiac ACE2 [1]. As with the case with diabetes, hypertensive subjects also exhibited reduced ACE2 levels in both kidney and heart compared to normotensive volunteers [48], probably due to Ang II-related mechanism [5]. Interestingly, renal ACE2 expression inversely correlates with hypertension [46,49]. Finally, intraperitoneal administration of recombinant human ACE2 (2 mg/kg⁻^1^/d⁻^1^) prevented Ang II-induced hypertension, renal oxidative stress and tubulointerstitial fibrosis in a mice model [50], suggesting ACE2 as a potential therapeutic treatment for cardiovascular and renal diseases.

Collectively, these results suggest a beneficial role of ACE2 against the deleterious structural and functional changes characterizing certain kidney diseases on one hand, and that ACE2 might be an important therapeutic target during these clinical settings on the other [20]. The upregulation of renal ACE2 in these disease states may represent a compensatory response aimed at counterbalancing the adverse effects of the classic RAAS system. This concept is further supported by the observation that pediatric patients with end-stage renal disease exhibited dramatic increases in plasma Ang(1-7) levels [51], which along with the MasR represent a valuable compensatory arm in the diseased kidney. In summary, ACE2/Ang 1-7/MasR axis exerts preferred antioxidative stress and antifibrotic effects in the face of the ACE/Ang II/AT1R devastating machinery.

## 4. COVID-19

This novel human-pathogenic coronavirus, named by the World Health Organization (WHO) “COVID-19”, belongs to the Betacoronaviruses (βCoVs) of the coronavirus family 2 [52] that caused the Severe Acute Respiratory Syndrome (SARS-CoV-1) and the Middle East Respiratory Syndrome (MERS-CoV) epidemics in 2003 and 2012, respectively. Coronaviruses are a family of single-stranded enveloped RNA viruses that are divided into four major genera (α, β, ϒ, 𝛿) [53]. Evolutionary analyses have shown that bats and rodents are the source of most βCoVs, and the supposed intermediate host of COVID-19 is probably the Pangolin. Indeed, metagenomics sequencing of the pangolin associated coronavirus shares a 99% sequence similarity with the strain of COVID-19 related to a specific site known as the receptor-binding domain in ACE2 that is essential for cell entry [54]. The outbreak emerged in Wuhan, China during December 2019, manifested as a mysterious atypical viral pneumonia 1 [55], and eventually progressed to a WHO-declared pandemic by March, 2020 [56]. Covid-19 has rapid person-to-person transmission with an estimated basic reproductive number (R_0_) of 2.2 [55,57]. One of the main reasons for this high R_0_ may be related to the variability in symptoms and severity of the disease, which range from a totally asymptomatic mild disease among the majority of cases (more than 80%) to severely symptomatic [58]. The mean incubation period is of 5.2 days (95% confidence interval (CI), 4.1 to 7.0), with the 95th percentile of the distribution at 12.5 days [57], thus supporting the 14 days quarantine.

Aerobiology teaches us that respiratory droplets are large (>5 μm) and remain in the air for shorter times and distances (<1 m) [59,60] compared to the airborne transmission by aerosolized droplets, which are smaller (<5 μm) and can remain in the air for longer time periods (days-week) and travel distances up to 20 m. Co-occurrence of rotes of transmission [61] depend on the temperature and the relative humidity. Lower temperatures (7–8 °C) favor for airborne virus survival, while relative humidity is an important factor in the viability of both the airborne and droplet viral transmission [62,63,64]. Covid-19 is mainly a droplet contact disease, emanating from direct close contact with infected people, by contact with surfaces in their immediate environment or by contact with contaminated by feces and urine with transfer to respiratory mucus membranes, and lastly by airborne transmission in specific circumstances [65,66]. Of note is that transmission is likely to occur only when the lower respiratory tract disease develops [58].

All the population is generally susceptible, but a difference from many other pandemic infectious diseases in the past such as malaria and the Spanish flu, where young children were mostly affected, in this present pandemic the elderly are the most commonly targeted (≥70 years old) for serious disease. Indeed, the median age of COVID-19 infection is 59 years, with a male predominance compared to female (ratio 2:1) [57,67].

Although the case fatality rate (CFR) of COVID-19 is ~6–7%, lower than in the outbreak of the SARS epidemic in 2003, with a case fatality rate (CFR) of 9.5%, and much lower than the MERS epidemic in 2012, with CFR of 34.5%, COVID-19 has caused a higher number of deaths than both prior epidemics together [68]. The CFR of COVID-19, varies according to age, being highest in the elderly with rapid progression and death, [69] CFR also depends on comorbidities, and ranges from 0.9% in healthy asymptomatic infected patients to 10.5% in patients with preexisting cardiovascular disease, 7.3% in diabetics, 6.3% in chronic obstructive pulmonary disease (COPD), 6% in hypertensive patients and 5.3% in cancer patients [52]. Acute kidney injury (AKI) is an independent risk factor for patients’ in-hospital mortality [70,71], and if acute renal failure (ARF) (AKI requiring RRT) develops, this carries the highest risk of mortality (60–90%) [55,70,72].

## 5. The Cardinal Clinical Manifestations

Infected patients with COVID-19 exhibit a wide spectrum of symptoms and severity, ranging from totally asymptomatic carriers, or those with mild constitutional symptoms such as fever (43%-98%), myalgia (11–15%), and upper (rhinorrhea and sore throat) and lower respiratory symptoms (dyspnea 35–64%; chest tightness; cough (68–82%), to severe symptoms such as bilateral interstitial pneumonia, acute respiratory distress syndrome (ARDS), septic shock with multiorgan failure (M.O.F.) due to cytokine release storm, acute cardiac damage, AKI and death. Less commonly, the presenting symptoms are related to the gastrointestinal tract, such as diarrhea, nausea or vomiting. In some patients, especially the elderly, “silent hypoxemia” and respiratory failure without dyspnea are observed [73].

Paradoxically, dialysis patients with COVID-19 had milder clinical disease according to data from a single hemodialysis center from Wuhan, in which 37 cases of 230 hemodialysis patients (16%) were diagnosed with COVID-19 infection. Six of them (16.2%) died from cardio-cerebrovascular diseases, and hyperkalemia rather than pneumonia-related complications [74].

Severe pneumonia of viral or bacterial origin, sepsis, and severe pneumonia or M.O.F-COVID-19-related infections are characterized by a transitory hypercoagubilty state along with an inflammatory state. In fact, early changes of clotting, platelet activation and artery dysfunction are common manifestations of COVID-19, as evident by cardiovascular complications such as myocardial infarction (MI) and ischemic stroke (CVA) as well as pulmonary vascular clots. Indeed, in COVID-19 infection there is an imbalance between the activation and inhibition of the coagulation system, favoring the activation one. In this context, an increase in levels of D-Dimers, fibrinogen, prolonged international normalized ratio (INR), enhanced levels of the prothrombin fragment F1 + 2 (marker of thrombin generation) along with impaired activation of the anticoagulant protein “C”, are observed. Platelets activation is also involved and results in overproduction of platelet thromboxane (Tx) B2 [75], a known risk factor for MI, as platelet TxB2 was independently associated with the occurrence of MI. Finally, patients with sepsis or severe pneumonia also develop impaired artery dilatation by an unknown mechanism, supposed to be secondary to upregulation of the NOX-2 enzyme responsible for reactive oxidant species (ROS) production, favoring clotting, platelets activation and impaired arterial dilatation [76]. Data regarding clotting changes in COVID-19 infection are still incomplete. Neither the increase in D-dimer nor the prolonged INR and thrombocytopenia are specific for clotting activation [77], but rather indicate a poor prognosis and survival. Whether or not to treat patients with severe COVID-19 infection with anticoagulation therapy such as aspirin or low molecular weight heparin (LMWH) is still under investigation. Nevertheless, Falcone et al. showed lower mortality in patients with community-acquired pneumonia that were treated with low dose aspirin compared to those who were not (4.9% vs. 23.4%; *p* <0.001; HR 2.07) where less nonfatal cardiovascular events were observed (4.9% vs. 8.3%; odds ratio, 1.77; *p* = 0.040) [78]. Also Tang et al. showed that treatment with LMWH appears to be associated with better prognosis among severe COVID-19-infected patients with sepsis-induced coagulopathy (SIC) [79].

## 6. Laboratory Tests

Positive patients for COVID-19 have a normal white blood cells (WBC) count but lymphopenia (the lymphocytes express ACE2 receptors) is commonly seen (~80%) [72,80]. A neutrophil/lymphocyte ratio >3 has poor prognosis. Mild thrombocytopenia is common and correlates with poor prognosis [69]. High serum levels of troponin I (strong prognostic indicator for mortality), CRP and D-Dimer correlate with disease severity and prognosis [69,74]. Procalcitonin, if elevated, may suggest a superimposed bacterial infection [72]. Proteinuria and hematuria were found in approximately 40% of admitted patients [71]. Paradoxically, dialysis patients with COVID-19 had less lymphopenia and lower serum levels of inflammatory cytokines [74]. Computed tomography (CT) scan of the kidneys showed reduced density, suggestive of inflammation and edema [70].

## 7. Kidney Involvement

The incidence of AKI in COVID-19-infected patients defined according to the kidney disease improving global outcome (KDIGO) criteria [70] is 3–9%, which is relatively low compared to the previous SARS and MERS-CoV infections (5%-15%). However, recent publications reported a higher frequency of kidney involvement [70]. A large consecutive cohort study [70] evaluated the association between markers of abnormal kidney function and death in 701 COVID-19 patients with a median age of 63 years, of whom 2% had chronic kidney disease (CKD). On admission, 43.9% of patients had proteinuria and 26.7% had hematuria. The incidence of elevated serum creatinine (Scr), blood urea nitrogen (BUN) and reduction in estimated glomerular filtration (eGFR) under 60 mL/min/1.73 m^2^ was over 14%. During hospitalization, 5.1% of patients developed AKI and as expected, the incidence was significantly higher in patients with high baseline Scr on admission (11.9%) than in patients with normal baseline Scr (4.0%). Moreover, patients with elevated baseline Scr experienced earlier (on day two) and more severe AKI (stage “3”: 4%) than patients with normal baseline Scr (stage “1”: 1.9%). In comparison to patients with normal baseline Scr (0.77 mg/dL), subjects with high baseline Scr at admission (1.91 mg/dL), were older (72 vs. 61 y), of male sex (72.7 vs. 49%), with more severe disease (52.5 vs. 40.7%), more comorbidity (60 vs. 39.6%), CKD (9% vs. 0.8%), had a higher leukocyte count, lower lymphocyte count and platelet count, more coagulation test abnormality, higher D-dimer levels, increased procalcitonin and high level of aspartate aminotransferase and lactose dehydrogenase. CT scan of the kidneys showed reduced density, suggestive of inflammation and edema [70] There was a dose-dependent relationship between AKI stages and in hospital death, with an excess risk of mortality by at least four times among those with stage 3 AKI (33.7 vs. 13.2%; *p*-value < 0.001).

Even after adjusting for age, sex, disease severity, comorbidities, lymphocyte count, peak serum creatinine >1.5 mg/dL and AKI over stage 2, elevated baseline Scr, BUN, proteinuria and hematuria of any degree were all associated with in-hospital death. Of note, 33/701 (4.7%) of the patients on admission were on RAAS inhibitor treatment and none of them developed AKI, and even when the use of these medications during hospitalization increased to 6.2%, there were no new AKI cases.

In an additional single-centered, retrospective and observational study, 52 of 710 critically ill patients with COVID-19 pneumonia were admitted to the intensive care unit (ICU). Fifteen out of 52 patients (29%) developed AKI, 9/52 (29%) were treated with renal replacement therapy (RRT) and 8/9 died. The total death at 28 days was 61.5% [80]. In contrast, a recent study enrolled 116 COVID-19 patients, 4.3% of them were on chronic RRT [81]. Specifically, mild and severe pneumonia were developed in 59/116 (50.8%) and 46/116 (39.7%) respectively, while the remaining 11/116 (9.5%) developed ARDS and transferred to ICU. Remarkably, patients without CKD that were infected with SARS-CoV-2 did not develop AKI. Moreover, CKD patients who were undergone regular CRRT and were infected with SARS-CoV-2 remained in a completely stable state, and all of them survived. It is worthy of mention that all seven patients (6.3%) who died were in the ICU, with ARDS, over 60 years old and with comorbidities but, interestingly, none of them had CKD or developed AKI. A retrospective analysis of the clinical data on kidney function from 85 Chinese infected patients with COVID-19 showed that 27.06% (23/85) developed AKI [82]. Another retrospective, single-center study [55] of 138 hospitalized patients with COVID-19-related pneumonia, with median age was 56 years, showed that 3.6% (5/138) developed AKI (3/5 patients were admitted in ICU). Guan et al. presented the data of clinical characteristics of 1099 infected patients from 552 hospitals in China. Renal function showed that the patients’ number of Scr ≥1.5 mg/dL was 12/752 (1.6%) [72].

A recent, large, consecutive cohort study of hospitalized patients with AKI-COVID -19 induced in metropolitan New York hospitals showed a higher incidence of AKI: 1993 of 5549 (36.6%) in admitted patients (probably due to higher rate of comorbidities and severity of respiratory disease), the majority of whom were classified as stage “1” (46.5%) and fewer stage “2” (22.4%) and stage “3” (31.1%). As expected, most of the patients who were mechanically ventilated developed AKI (89.7%), stage “3”, and required RRT (66.7%) as early as one day after admission and within 24 h of intubation (52.5%). The major risk factors for AKI were the need for both mechanical ventilation and vasopressor drugs [83]. As in other diseases complicated with AKI, COVID-19 with AKI carried a poor prognosis. Indeed, the mortality rate in COVID-19 patients who developed AKI was 35%, and in patients who required RRT the mortality rate was even higher (55%) [83].

In summary, according to the American Society of Nephrology (ASN) the incidence of AKI in patients with COVID-19 is variable where AKI occurs in approximately 15% of all ICU admissions. In fact, according to above-listed studies, the incidence of AKI in COVID-19 infected patients is widely different, ranging from totally absence to a high incidence rate (~40%), so we should wait for more data to have a clearer picture.

## 8. The Etiology of COVID-19-Induced AKI

Patients at risk for AKI are elderly (≥60 years; 65.22% vs. 24.19%, *p* <0.001) or carry comorbidities (69.57% vs. 11.29%, *p* <0.001), such as hypertension (39.13% vs. 12.90%, *p* = 0.0007) and coronary heart disease (21.74% vs. 4.84%, *p* = 0.018), suggesting renal function impairment is relatively common in COVID-19 patients [82]. In concordance with this study, a new large cohort study [83] showed that respiratory failure with mechanical ventilation and the use of vasopressor drugs are the most important risk factors for AKI development (OR 10.7; 4.53 respectively).

According to postmortem autopsies that were performed in six patients [82], the main kidney damage is at the tubulointerstitium with varying degrees of acute tubular necrosis, luminal brush border sloughing and vacuole degeneration, and varying degrees of lymphocyte infiltration. Viral infection associated-syncytia were also observed in three cases. At the glomerular level, dilated capillary vessels were observed, but no severe glomerular injury was evident. To explore the mechanisms involved in the pathogenesis of tubule-interstitial damage, immunohistochemistry staining was performed and showed the viral nucleocapsid protein (NP) antigen in kidney tubules, strong presence of CD68+ macrophages and complement C5b-9 depositions in the tubulointerstitium that can further accelerate and amplify kidney damage. These findings suggest that the tubular damage is caused by either a direct cytotoxicity of COVID-19 or by immune-mediates tubule pathogenesis [82]. In this context, analysis of postmortem autopsy data of three infected patients revealed degeneration and necrosis of parenchymal cells, formation of hyaline thrombus in small vessels and pathological changes of chronic diseases were observed in various organs and tissues including the kidneys [84]. Recently, COVID-19 virus particles were isolated from the urine of COVID-19 patients, suggesting that kidney-originated viral particles may enter the urine through glomerular filtration [52].

As mentioned above, ACE-2 is highly expressed in the brush border of proximal tubular cells but much less in podocytes [4,17,18]. There is no evidence that ACE-2 is present in mesangial or in glomerular endothelial cells [17] and this may explain the absence or the mild glomerular involvement.

However, the susceptibility of black patients to COVID-19, including renal manifestation, suggests involvement of apolipoprotein L1 (APOL1) variants, which are more common in those of African descent, in this phenomenon. Indeed, three case reports demonstrated collapsing glomerulopathy in patients with SARS-CoV-2 infection, where all of them were of African origin and carried APOL1 G1 risk allele homozygosity. [85,86,87]. In line with these observations, a more comprehensive study by Wu et al. [88] demonstrated that black patients (*n* = 6) with COVID-19 presenting with AKI and de novo nephrotic-range proteinuria, exhibited collapsing glomerulopathy, extensive foot process effacement and focal/diffuse acute tubular injury in kidney biopsy specimens. In addition, three patients displayed endothelial reticular aggregates. However, there was no evidence of viral particles or SARS-CoV-2 RNA. Interestingly, all six patients had an APOL1 high-risk genotype (G1 risk allele). Five patients needed dialysis (two of whom died); one partially recovered without dialysis. These findings suggest that black individuals with an APOL1 high-risk genotype and severe acute respiratory syndrome coronavirus 2 infection are at increased risk for experiencing an aggressive form of kidney disease associated with high rates of kidney failure.

## 9. Treatment

Currently, there is no specific pharmacological treatment for COVID-19. Fortunately, the majority of the infected patients present with mild symptoms and are treated accordingly, with only supportive treatment. It is of utmost importance to avoid fluid resuscitation unless there is a real need (hypovolemic patients). Noteworthy, the main cause of death from COVID-19 is ARDS, which may be exacerbated by fluid administration (worsen hypoxemia). In symptomatic patients, few studies have shown that the use of the antimalarial drug, chloroquine, exerted beneficial antiviral effects [89]. A recent study of more than 100 patients with COVID-19 pneumonia showed the superiority of chloroquine phosphate over the control treatment in inhibiting pneumonia exacerbation, promoting seroconversion and shortening the disease course [90,91]. An acidic endosomal pH is critical for COVID-19 processing and internalization, an obligatory step before starting the replicative phase. Indeed, chloroquine phosphate has the ability to increase the endosomal pH due to its accumulation in the endosomes, interfering with the glycosylation of cellular receptors and fusion process between the COVID-19 endosome [90,91,92]. However, other studies reported an adverse impact of these drugs on COVID-19 patients’ outcome [93,94,95,96], intensifying the debate concerning the utilization of hydroxyl chloroquine during this disease state. In this regard, a few small trials did not show any benefit of using chloroquine in COVID-19-infected patients [93,94,95,96]. Currently, there are twenty-three ongoing trials that should shed light on the role of chloroquine phosphate treatment in patients infected with COVID-19. Meanwhile, both the Infectious Diseases Society of America (IDSA) [97] and the National Institute of Health (NIH) [98] recommend the use of hydroxychloroquine therapy only in the context of a clinical trial. However, a recent study at a large medical center in New York City that examined the association between hydroxychloroquine use and intubation or death of patients with Covid-19, revealed that hydroxychloroquine administration was not associated with either a greatly lowered or an increased risk of the composite end-point of intubation or death [99]. However, most recent meta-analysis studies demonstrated that hydroxychloroquine administration might cause harm in COVID-19 patients [100,101].

No specific antiviral therapy has been proven as a treatment for COVID-19 infection in humans, and multiple randomized controlled trials (RCTs) are ongoing. Remdesivir might be an potential antiviral, tested in vitro and in animals with MERS [102]. Indeed, the most recent clinical study on 1059 patients (538 assigned to remdesivir and 521 to placebo) demonstrated that remdesivir was superior to placebo in shortening the time to recovery in adults hospitalized with Covid-19, with evidence of lower respiratory tract involvement [103].

Lopinavir/ritonavir a combination of antiviral agents used in treatment of HIV, is being used in COVID-19 with some success. Lopinavir/ritonavir, which blocks viral replication, appears to work synergistically with ribavirin. Available human data showed that poor clinical outcomes (ARDS or death) were lower in the treatment group (2.4% vs. 29%) and caused significant reduction in viral load in patients with SARS [104]. Another study showed reduced mortality (2.3% versus 16%) [105]. In a cohort study describing 16 COVID-19 patients in Singapore, in which six of them suffered from hypoxemia, five patients were treated with low dose lopinavir/ritonavir [74]. Among them, two patients deteriorated and had persistent nasopharyngeal virus carriage. Unfortunately, end-stage renal disease (ESRD) or dialysis patients were not included. Lopinavir/ritonavir is currently under investigation within multiple RCTs in China. Tocilizumab, an antagonist of the IL-6 receptor, has been used to treat patients with IL-6 cytokine storm with good results [106]. Fortunately, dialysis patients did not develop this cytokine storm [74].

The use of glucocorticoids is still controversial, at least in acute respiratory distress syndrome (ARDS), due to viral infection. Acute lung injury and ARDS are partly caused by host immune responses [107]. Corticosteroids suppress lung inflammation but also inhibit immune responses and pathogen clearance. In a retrospective observational study reporting on 309 adults who were critically ill with MERS, patients who were given corticosteroids were more likely to require mechanical ventilation, vasopressors, renal replacement therapy and impaired clearance of viral RNA from respiratory tract and blood, and treatment was not associated with a difference in 90-day mortality [57,108]. However, some studies advocate for the use of a low dose and short regimen of corticosteroids in severely ill COVID-19 patients [109]. In line with these findings, a recent study in a large population of hospitalized patients with Covid-19 demonstrated that the use of dexamethasone resulted in lower 28-day mortality among those who were receiving either invasive mechanical ventilation or oxygen alone at randomization, but not among those receiving no respiratory support [110]. Based on this encouraging report, many medical centers in the world adopted administration of dexamethasone at a dose of 6 mg once daily for up to 10 days as a routine therapeutic protocol for patients with severe COVID-19. Collectively, the WHO recommended the use of systemic corticosteroids for the treatment of patients with severe and critical COVID-19, but not in patients with nonsevere COVID-19, as the treatment brought no benefits and could even prove harmful [57,111].

An additional therapeutic approach relies on the early use of hyper immunoglobulin therapy from convalescent patients infected with SARS and other viral etiologies, as it accelerates recovery and reduces mortality of these patients [112]. There are two ongoing trials on the efficacy of convalescent plasma in patients with COVID-19 [113].

## 10. Renal Replacement Therapy

The evolving history of AKI in COVID-19 patients without previous CKD revealed that most patients returned to baseline kidney function without the need for chronic renal replacement therapy (RRT). Similarly, in patients with AKI superimposed on CKD, most of them recovered completely without further deterioration in kidney functions [114]. The indications for starting RRT in COVID-19-induced AKI are similar to those of other etiologies. CRRT has been successfully applied in the treatment of SARS, MERS, and sepsis, and so it is recommended as a rescue treatment for preserving kidney function in COVID-19-induced AKI cases [115,116,117]. These studies showed that treatment with high-volume hemofiltration (HVHF) at a dose of 6 L/hr of sepsis-induced ARF significantly reduced IL-6 levels (*p* = 0.025) and the overall organ failure assessment (SOFA) score was improved after one week, suggesting the important role of CRRT in patients who develop ARF, though further studies are needed. One of the pathophysiologic hypotheses involved in COVID-19-induced ARF is the cytokines release storm, and treating these patients with CRRT therapies by haemofiltration and haemodiafiltration may lead to the resolution of multiple organ dysfunction syndrome (MODS) [58].

Another modality of treatment is the use of the adsorption technique hemoadsorption that places sorbents in direct contact with blood via an extracorporeal circuit for solute removal (by both diffusion and convection). This approach has been utilized for years [118,119,120], especially in intoxication (removal of exogenous toxins), and recently its application has been expanded to other conditions, either acute or chronic, such as inflammatory conditions, or chronic uremic symptoms and autoimmune disease. It was shown in two randomized controlled trials (RCT) that using an HA-330 cartridge in patients with sepsis or acute lung injury due to sepsis significantly improved survival (reduced both ICU and in hospital mortality), stay length in ICU, hemodynamic parameters (less vasopressors use) and significantly reduced the proinflammatory cytokines, suggesting its safety [121,122]. The aim of this therapy is to remove as much, and as early as possible, the excess of circulating cytokines, with consequent singular benefits in terms of hemodynamic parameters and recovery of multiple organs dysfunction [123]. One of the major concerns is the nonselectivity in removal of both circulating pro and anti-inflammatory cytokines, with deleterious consequences. In fact, a low-level TNF response seems to be requisite for the host defense against infection and, vice versa, high levels are needed to be regulated by an anti-inflammatory feedback because of the high risk of organ dysfunction and death. Unfortunately, a loss of this proper immunologic balance, as in sepsis, can induce a state of acquired immunodeficiency called immunoparalysis, due to an excessive anti-inflammatory response, and exposes the host to further infections and death [124,125,126]. A proper immunologic balance is vital and lifesaving. CRRT is a continuously acting therapy that helps removing pro and anti-inflammatory cytokines in a non-selective way; a concept known as “peak concentration hypothesis” [127] as suggested by human studies on sepsis in which circulating cytokines with the highest concentration are removed in the highest amounts, thus reducing the relative excess of active substances [128,129]. Further studies are still needed, not only regarding efficacy but also safety, especially with respect to potential hypotension and bleeding tendency [120].

Finally, an additional potential adjunctive therapy is the lectin affinity plasmapheresis (LAP) that was used in various viremia caused by enveloped viruses such as MER-CoV, SARS-CoV, HIV and Marburg virus (MARV), and should be considered for the therapeutic armamentarium of COVID-19-infected patients. Briefly, blood runs into a plasma filter that has high affinity to the glycoprotein (GP) on the on the surface of enveloped viruses and to the soluble form of GP (sGP) secreted from the infected cells into the circulation [130]. Volchkova et al. [131] showed that the sGP produced by the MERS-CoV shares over 90% of its sequence with the N-terminal region of GP1,2 (on the surface of enveloped MER-CoV), and acts as a decoy in neutralizing antibodies against GP1,2 (cross-reaction), thus allowing virus survival within the host [132]. This shift towards non-neutralizing antibody formation, a novel host immune evasion mechanism is called the “antigenic subversion” phenomenon [133]. In addition, the sGP have been shown to induce a massive release of cytokines and to increase vascular permeability [134]. A time-dependent clearance of both the virus (up to 80% within 3 h), the soluble GP and other immunosuppressive viral fragments (up to 70% within 3 h) by LAP results in decreasing inflammation and restores the immunity system hemostasis by releasing more antibodies to fight the offensive virus, thus improving the survival rate [135,136,137,138,139,140]. This therapy warrants further evaluation in RCT.

## 11. Kidney Transplanted Patients

The risk of infection in kidney-transplanted patients is similar to the general population. There are no data regarding disease severity once infected, but probably, as with other viruses, the disease is more severe in view of low immunity due to the use of immunosuppressive drugs [141,142]. As expected, there are no guidelines regarding the management of kidney transplant immunosuppression in infected patients requiring hospitalization. One approach is classified according to two indications: age (< or >60 years old) and absence or presence of pulmonary infiltrate [141]. Another approach is to stop all the antirejection medications except steroids for few days till recovery, or at least till improvement of hemodynamic or medical conditions. Recently, five cases of kidney transplant patients with COVID-19 infection were published [143,144,145,146], which shed light on the course and prognosis of these high-risk patients. The median age was of 56.5 (49–75) and the presenting symptoms did not differ from those of the general population, fever being the main presenting symptom. All of them developed unilateral pneumonia, except one who had bilateral pneumonia, but none necessitated mechanical ventilation. Only one had mild AKI (stage 1) at admission due to dehydration, but none experienced graft rejection or necessitated RRT. The antirejection therapy was discontinued immediately, except the steroid therapy that, on the contrary, was increased to a stress dose or more (20–80 mg/d methylprednisolone) for several days to two weeks to prevent acute rejection, adrenal crises and, at the same time, treat the inflammatory component of the infection itself, including lung interstitial exudation. The antirejection therapy was resumed gradually after symptoms declined.

Antiretroviral therapy (Lopinavir/Ritonavir, Darunavir, Cobicistat and Interferon β1), empirical antibiotics, intravenous immunoglobulin to prevent secondary infections, and hydroxychloroquine and colchicine treatment were used in transplanted patients. The latter prevents NLRP3 inflammasome assembly [147] with consequent reduction of the interleukins release including IL-6 and others, and is used whenever the anti-IL-6 receptor mAb tocilizumab is unavailable. Close monitoring and dose adjustment of the calcineurin inhibitors (CNI) or mammalian target of rapamycin signal inhibitors (imTOR) levels is obligatory in view of the interaction with these drugs. It is worthy of mention that 2/5 patients were on chronic ARBs treatment and continued to receive it throughout their disease without further complications. Fortunately, full recovery was the common feature in all the five patients. In summary, the course of COVID-19 in kidney transplanted patients did not significantly differ from that of the general population. However, a recent small study by [148] showed data on 36-adult kidney-transplant recipients infected with Covid-19, in which 27 of them (75%) had received a cadaveric kidney and 26 of them (72%) were male, with median age of 60 years. Fourteen recipients (39%) were black, and 15 recipients (42%) were Hispanic; 34 recipients (94%) had hypertension, 25 (69%) had diabetes mellitus, 13 (36%) had a history of smoking tobacco or were current smokers and six (17%) had heart disease. Thirty-five of them (97%) were receiving tacrolimus, 34 (94%) were on prednisone, and 31 (86%) were receiving mycophenolate mofetil. The presenting symptom was fever in 21 patients (58%), followed by cough in 19 patients (54%) and diarrhea in eight patients (22%). Twenty- eight patients (78%) were admitted to the hospital and eight patients (22%) were stable and discharged at home for quarantine. Twenty-seven of the hospitalized patients (96%) had viral pneumonia, Eleven of twenty-eight (39%) were mechanically ventilated and 6/28 (21%) were treated with RRT. Ten of the thirty-six kidney transplant recipients (28%) and 7/11 of the intubated patients (64%) died. Two of the eight patients (28%) who were monitored at home died; both were recent kidney-transplant recipients who had received antithymocyte globulin (ATG) within the previous five weeks. Twenty-two (79%) of the hospitalized patients were lymphopenic, 12 (43%) had thrombocytopenia, 19 (68%) had low CD3 cell counts, 20 (71%) had low CD4 cell counts, and eight (29%) had low CD8 cell counts. Inflammatory markers, such as ferritin were high in 10 patients (36%), CRP was high in 13 patients (46%), high procalcitonin levels in 12 patients (43%) and high d-dimer levels in 16 patients (57%). An additional study which included 20 kidney transplanted patients admitted for COVID-19 pneumonia followed up for seven days revealed that 87% of them exhibited radiological progression, where 73% developed hypoxemia that required oxygen therapy. Six patients developed ARF. Overall, five kidney transplant recipients died after a median period of 15 days [149]. Collectively, these results demonstrate that kidney-transplant recipients appear to be at particularly high risk for critical Covid-19 illness due to chronic immunosuppression and coexisting conditions.

Immunosuppressive therapy rearrangement is mandatory in view of the very low levels of CD3, CD4 and CD8 cell counts. In fact, withdrawal of antimetabolite was done in 24 of 28 patients (86%) and tacrolimus was withheld in six of the 28 severely ill patients (21%). Hydroxychloroquine was administered to 24 of these 28 patients (86%). Apixaban was administered to patients with D-dimer levels higher than 3.0 μg/mL. Six severely ill patients received the CCR5 antagonist leronlimab (a humanized IgG4 mAb that blocks CCR5, a cellular receptor used in HIV infected patients), and two received tocilizumab. However, only one patient, who had the lowest interleukin-6 level, remained in stable condition without intubation. In sum, this report showed a different clinical course and outcome. Specifically, a more rapid clinical progression was observed and a very high early mortality rate of 28% at three weeks was evident as compared with the reported 1% to 5% mortality among patients with Covid-19 in the general population in the United States. The factors underlying these differences in clinical course and outcome between the two published studies could be attributed to different protocols of immunosuppressive therapy withdrawal, different antiviral therapies, and may be due to the various number of transplanted patients included in each study.

## 12. Use of RAAS Blockers among COVID-19

The high rate of mortality among COVID-19 patients is still matter of challenge [69]. One hypothesis is that the use of angiotensin converting enzyme inhibitors (ACEi), angiotensin receptor blockers (ARBs) or aldosterone antagonists (MRA) [114,150,151] increases the patient risk of infectivity due to upregulation of ACE-2 receptors which serves as a gate entry into the cell where it replicates (Figure 2). The upregulation of ACE2 by ARBS was seen in some tissues such as kidney and heart (2–5 fold), but there is no direct evidence in other tissues such as the lungs [47,152,153,154]. It should be emphasized that SARS-CoV-2 entrance into the cell needs the participation of a serine protease TMPRSS2 (transmembrane protease, serine 2) that is not under the influence of renin–angiotensin system (RAS) [155]. The question is: does treatment with ACEi or ARBs predispose these patients at risk for COVID-19 infection and, consequently, should it be stopped prematurely?

There is no evidence to support discontinuation of such treatment according to current knowledge. First, at the lung level, downregulation of ACE-2 receptors facilitates neutrophil recruitment induced by endotoxins [156], exacerbates lung injury and increases mortality, perhaps due to in situ production of Ang II [157]. Animals studies have shown that in severe lung injury, ACE-2 receptors are deeply downregulated [158] and treatment with ARBs attenuates the lung injury in view of its anti-inflammatory properties, thus decreasing further deterioration to ARDS [156,159]. The proposed mechanism is attributed to the stimulatory effects of ARBs on ACE-2 expression/abundance. Support for this notion is derived from the work of Gu et al., who showed that intravenous infusion of recombinant ACE-2 into mice infected by respiratory syncytial infection virus (RSV), attenuated the severity of RSV-induced lung injury [157]. Recent human studies have investigated the effects of RAAS inhibition on ACE2 expression. In one study [160], IV infusion of ACE inhibitors into patients with ischemic heart disease did not affect Ang (1-7) production. Another study, conducted in hypertensive patients treated with captopril, showed Ang (1-7) levels were increased only after long-term therapy (>6 months) [161]. Further studies were done in patients with cardiac diseases and showed no changes in the soluble ACE-2 activity or urine shedding following treatment with either ACEi or ARBs [162,163,164,165]. In a longitudinal cohort study conducted in hypertensive Japanese patients, urinary ACE2 shedding was higher among patients treated with olmesartan, but was not observed with other medications from the same group (ARBs) or ACEi [166]. Recently, it was shown by Reddy et al. that in ARDS patients Ang I was increased but Ang (1–9), the precursor of Ang (1-7), concentration was reduced in the nonsurvivor group, suggesting a reduction in ACE2 activity [167]. Mizuiri et al. also showed that diabetic patients with overt nephropathy have high ACE/ACE2 ratio in kidneys (probably induced by prolonged hyperglycemia), suggesting a role in the progression of renal injury and the vulnerability of the kidneys in case of infections, including COVID-19 [35]. Lastly, in a recent cohort study of 5449 COVID-19-infected patients, 28% of them were on ACEi or were ARBs users, and there was no evidence of increase in AKI risk or higher mortality rate [83]. Secondly, treatment with ARBs increases the soluble form of ACE-2, albeit slightly (from 2% to 4%), by cleaving the ectodomain of the ACE-2 receptor via the action of the metalloproteinase ADAM17 (ACE-2 shedding). Doing that, more COVID-19 is captured in the circulation, preventing or reducing the cell COVID-19 infection rate [157]. Thirdly, ACE-2 is only one of multiple players involved in viral cell entry, which includes also TMPRRSS2, an essential serine protease not influenced by ACEi or ARBs, and once this enters the cell the replicative phase is also not affected by ACEi or ARBs. Fourth, not only the degree of expression, but also the biologic relevance of ACE2 may differ according to tissue and clinical state, suggesting that ACE-2 in the lung may have a different effect, protecting or attenuating the lung injury produced by COVID-19. Fifth, in view of the viral cardiotropism, there is no doubt of the benefits to ACEi and ARBs-treated patients in preventing acute cardiac injury during COVID-19 infection [168]. The actual recommendations for ACEi/ARBs use in COVID-19 patients are:Continue ARBs or ACE inhibitors in patients with indications.Do not start ARBs or ACE inhibitors solely for COVID-19.

There is an urgent need for a well-designed cohort prospective study to shed light on this issue.

## 13. Summary and Conclusions

COVID 19-is characterized by multi vital organ derangement as evidenced by respiratory distress, cardiac dysfunction and kidney injury. These adverse effects are largely attributed to the fact that SARS-CoV-2 utilizes ACE2 as invasion route into the cells accompanied by neutralization of the advantageous effects of this enzyme, especially in patients with background diseases such as diabetes, renal and heart failure. The fact that ACE2 is widely expressed in the renal tissue may sensitize this organ to the deleterious effects of SARS-CoV-2. This concept is supported by well-established evidence that the ACE2/Ang 1-7/MasR axis plays an important role in renal physiology and exerts nephroprotective effects against AKI of various etiologies. Considering the valuable actions of this axis, several treatments were exploited for therapeutic purposes including recombinant ACE2, ACE activators, MasR agonists and Ang 1-7. Most of these substances had already shown beneficial effects in experimental models of lung diseases, hypertension, and diabetes, but none in COVID-19. The latter may be of relevance, as these patients exhibit a high incidence of AKI, proteinuria and hematuria.

## Figures and Tables

**Figure 1 jcm-10-01216-f001:**
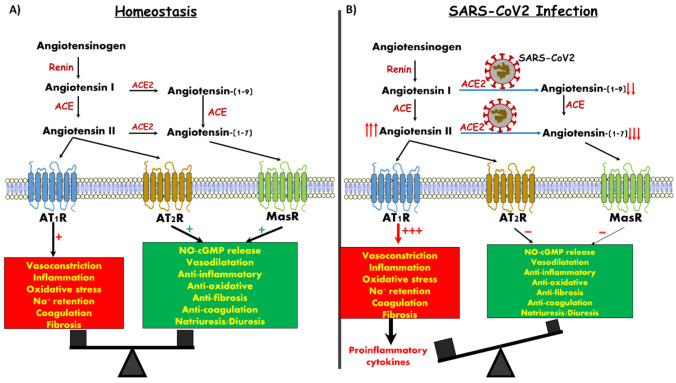
Angiotensin derivatives, their targets and downstream action. (**A**) Balanced impact of Ang II and Ang 1-7 on vascular tone and control of inflammation. (**B**) SARS-CoV-2 infection generates Ang 1-7 depletion, likely leading to unopposed vasoconstriction and inflammation.

**Figure 2 jcm-10-01216-f002:**
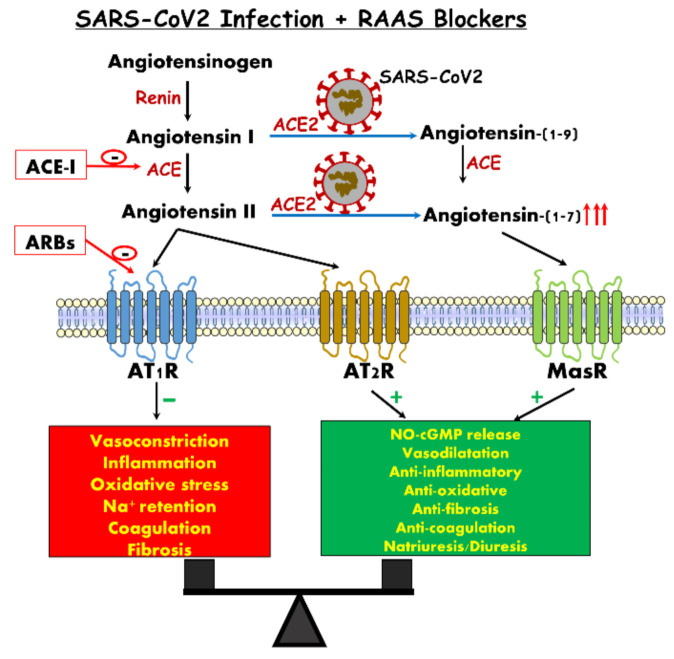
Concomitant renin–angiotensin system (RAS) inhibition with angiotensin converting enzyme (ACE) inhibitors or angiotensin II receptor blockers (ARBs) may restore the balance, with parallel suppression of signals mediated by angiotensin II receptor type 1 (AT1R) along augmentation of the angiotensin converting enzyme 2/ angiotensin 1-7/Mas Receptore (ACE2/Ang 1-7/MasR) axis.

## Data Availability

The original contributions presented in the study are included in the article; further inquiries can be directed to the corresponding author.

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
