# Peer review of "Renal Manifestations of Covid-19: Physiology and Pathophysiology"

_jcm, 2021, doi:10.3390/jcm10061216_

Round 1

Reviewer 1 Report

Dr Armaly and co-workers present a scholarly, comprehensive, and most timely overview of the renal angiotensin converting enzyme, its role and engagement in COVID-19, clinical and renal presentation of COVID-19 infection, therapeutic modalities and their use in COVID-19 patients, as well as analysis of COVID-19 in transplant patients. This overview is a welcome addition to the contemporary literature on the subject.

Minor issue - please complete the sentence on page 9, 1st para-: APOL1....

Author Response

Thank you for you kind feedback. APOL1 in page 9 was omitted, as it should not be there.

Reviewer 2 Report

In the current manuscript Armaly et al review implication of AKI in COVID-19.

The review is interesting and builds the story on a massive foundation of previous studies. This is the great strength of the review but also somewhat problematic as it's a bit heavy for the reader to penetrate. Maybe the review would be more accessible if the authors stripped down some of the text and focused on the take home messages. 

It is a solid review but a bit unfocused and too long, should be accepted when the take home messages (importance of the ACE2/Ang 1-7/MasR axis in COVID-19 AKI) have been clarified and a fair bit of text could be omitted under the sections:

Tretment

Renal replacement therapy

Kidney Transplanted Patients

Specific points:

  1. In the end of the abstract it says this is a mini review. Really it's not. Remove the word mini throughout the manuscript.
  2. After the introduction the first two head-lines are identical "Angiotensin Converting...". Typo?
  3. It's always nice with figures. But figure 1 could be a bit clearer and more intuitive on how the different factors interacts and affects each other.
  4. Figure 2 should be clarified. It looks very similar to figure 1 panel B. Clarify where the suggested actions to avoind AKI should be.   

Author Response

Thank you for your professional evaluation of our manuscript.

Native English speaker (Prof. Karl Skorecki) went through the manuscript and made minor grammar and style changes.

Comment: It is a solid review but a bit unfocused and too long, should be accepted when the take home messages (importance of the ACE2/Ang 1-7/MasR axis in COVID-19 AKI) have been clarified and a fair bit of text could be omitted under the sections:

Reply:Thank you for the note. Any comprehensive review on the pathophysiology of COVID-19-induced kidney injury must include the various therapeutic options as we did. We agree with you that this section is somehow long; therefore, we shortened this portion of the MS accordingly without affecting its content. See tracked changes file. Concerning the therapeutic approach to kidney transplanted patients, it should be emphasized that this topic is of special clinical relevance as these patients are at risk to develop infection in general and COVID19 in particular. Therefore, we choose to keep most of this sub-title.

Specific points:

  1. In the end of the abstract, it says this is a mini review. Really, it is not. Remove the word mini throughout the manuscript. Addressed.  
  2. After the introduction the first two head-lines are identical "Angiotensin Converting...". Typo? No; each title refers to different aspect: The first one “Angiotensin Converting Enzyme 2 and Kidney Function” whereas the second one “Angiotensin Converting Enzyme 2 and Kidney diseases”.
  3. It's always nice with figures. But figure 1 could be a bit clearer and more intuitive on how the different factors interacts and affects each other. The figures were slightly modified to stress the interaction between the beneficial and harmful arms of RAAS under normal and during COVID-19 conditions.
  4. Figure 2 should be clarified. It looks very similar to figure 1 panel B. Clarify where the suggested actions to avoid AKI should be.   

Indeed, Figure 2 is similar to Figure 1; however, it reflects the impact of RAAS inhibitors treatments on restoration of the balance between the beneficial and harmful arms of this system in patients with COVID-19. This figure includes the site of action of both ACEi and ARBs and their elimination of the deleterious effects of the ACE/AngII/AT1R axis. 

Reviewer 3 Report

In their manuscript, Armaly et al. review the biology by which SARS-CoV-2 virus enters cells, with emphasis on the kidney and renal disease that can ensue. Overall, the review is well-organized and thorough. The figures are very helpful. My most major concern is the grammar and writing throughout.

  • The middle of the abstract from “The SARS-CoV-2… to the Mas receptor” does not make sense. It appears like parts of the sentences were deleted then merged. This is is just one example, but careful reading and editing of the entire manuscript is required.
  • Page 4, last sentence on COVID-19 re. “gene sources”. I think gene is not the correct word.

Other suggestions:

  • Page 8, first sentence under TREATMENT needs to be modified, since now there are vaccines.
  • Pages 7 and 12 where authors describe results of various studies and have a lot of percentages, I would suggest putting this info in a table to make it easier for readers to take in the info. Difficult to digest all the info in sentences.

Author Response

Comment: Extensive editing of English language and style required

Reply: Native English speaker (Prof. Karl Skorecki) went through the manuscript and made minor grammar and style changes.

  • The middle of the abstract from “The SARS-CoV-2… to the Mas receptor” does not make sense. It appears like parts of the sentences were deleted then merged. This is is just one example, but careful reading and editing of the entire manuscript is required.

This specific sentence was rephrased (see abstract). In addition, we went through the MS and rephrased as needed. (See track changes)

  • Page 4, last sentence on COVID-19 re. “gene sources”. I think gene is not the correct word.

This sentence was corrected and “gene” was deleted, although the term gene sources was used by the cited work.

Other suggestions:

  • Page 8, first sentence under TREATMENT needs to be modified, since now there are vaccines.

The word “vaccines” was removed.

  • Pages 7 and 12 where authors describe results of various studies and have a lot of percentages, I would suggest putting this info in a table to make it easier for readers to take in the info. Difficult to digest all the info in sentences.

Indeed, these pages of the MS refer to “lot of percentages” concerning COVID19 manifestations. As these percentages are inconsistent among the various publications, it would be very difficult to prepare a concise and condensed table. We wish for your understanding.